# Radiation-driven acceleration in the expanding WR140 dust shell

Yinuo Han[1,2 ✉], Peter G. Tuthill[2], Ryan M. Lau[3,4] & Anthony Soulain[2,5]

The Wolf–Rayet (WR) binary system WR140 is a close (0.9–16.7 mas; ref.[1]) binary star consisting of an O5 primary and WC7 companion[2] and is known as the archetype of episodic dust-producing WRs. Dust in WR binaries is known to form in a confined stream originating from the collision of the two stellar winds, with orbital motion of the binary sculpting the large-scale dust structure into arcs as dust is swept radially outwards. It is understood that sensitive conditions required for dust production in WR140 are only met around periastron when the two stars are sufficiently close[2–4]. Here we present multiepoch imagery of the circumstellar dust shell of WR140. We constructed geometric models that closely trace the expansion of the intricately structured dust plume, showing that complex effects induced by orbital modulation may result in a 'Goldilocks zone' for dust production. We find that the expansion of the dust plume cannot be reproduced under the assumption of a simple uniform-speed outflow, finding instead the dust to be accelerating. This constitutes a direct kinematic record of dust motion under acceleration by radiation pressure and further highlights the complexity of the physical conditions in colliding-wind binaries.

The WR140 was observed on six occasions between 2001 and 2017 with the near-infrared cameras NIRC and NIRC2 at the Keck Observatory. The near-infrared imagery, displayed in Fig. 1, spans orbital phases between $\phi = 0.043$ and 0.592, clearly showing an expanding dust shell with an evolving apparent morphology. Although the images span two orbital cycles, features appear to be consistent, implying a high degree of cycle-to-cycle replication of the same underlying morphology. Prominent structures at earlier orbital phases ($\leq 0.111$) include an eastern and western dust arc. As the dust shell expands, these structures give rise to an 'eastern arm' and a prominent 'southern bar' at later orbital phases, following the nomenclature of structures identified in previous imaging[4].

The well-resolved detailed form of the plume motivates the construction of a geometric model to explain the structural variation and expansion over time. We modelled the geometry of the dust plume as a linearly expanding spiral based on the 'pinwheel mechanism'[5,6]. We assumed that dust production turns on and off episodically as the binary approaches, and departs the region near periastron. The wind-collision region can be approximated as the surface of a cone at large distances from the stars where the velocity of the compressed wind has reached its asymptotic value[7]. Dust assumed to form on this conical interface between the WR star and the O-star primary subsequently expands radially outwards as the binary continues in its orbit. Originally modelled on WR104 (ref.[5]), such pinwheel models have been shown to accurately reproduce dust structures in several Wolf–Rayet (WR) binaries including Apep[8] and WR112 (ref.[9]).

The orbital parameters of WR140 are well constrained[1,10] and form the basis for generation of the geometric dust plume model. By fitting to the location and geometry of dust structures across the multiple epochs of observations, our model suggests that the half-opening angle of the conical shock front is $\theta_w = 40 \pm 5°$. This value is in close agreement with that estimated by Fahed et al.[11] ($42 \pm 3°$), which is derived by fitting a cone model[12] to the 5696A C III emission line, and Williams et al.[13] ($34 \pm 1°$) by modelling the 10830 He I subpeak. The $\theta_w$ value estimated in this study implies a momentum ratio between the O and WR star of $\eta = 0.043^{+0.021}_{-0.015}$ assuming radiative postshock conditions[7,14], which is a larger value than that expected from the mass-loss rate and wind speed of the stars (Extended Data Table 2). Upon fitting to the turn-on and turn-off values independently, we find that dust production occurs over a period of $0.7^{+0.3}_{-0.1}$ yr centred at the periastron passage, with key parameters of the dust production thresholds displayed in Extended Data Table 3.

An image simulated under this geometric model at $\phi = 0.592$ is displayed in Fig. 2a. When comparing with the corresponding observations (Fig. 1), this model successfully reproduces a number of prominent features, with very accurate registration between the structural edges in the model and the data. The eastern arm represents a segment of the ellipse that corresponds to the earliest dust produced in the dust production episode (when dust production turned on), whereas the southern bar corresponds to the most recently produced dust in the episode (just before dust production turned off).

However, not all predicted features are apparent in the data. Although the geometric model is primarily designed to reproduce structural features, its physical interpretation implies a pinwheel system producing isothermal, optically thin dust at a constant rate. The clear absence of structures predicted by the geometric model, most noticeably dust features to the north and west (an outer-western arc) and below the southern bar, cannot be explained by simple density variations due

[1]Institute of Astronomy, University of Cambridge, Cambridge, UK. [2]Sydney Institute for Astronomy, School of Physics, The University of Sydney, Sydney, New South Wales, Australia. [3]NSF's NOIR Lab, Tucson, AZ, USA. [4]Institute of Space & Astronautical Science, Japan Aerospace Exploration Agency, Sagamihara, Japan. [5]Université Grenoble Alpes, CNRS, IPAG, Grenoble, France. ✉e-mail: yinuo.han@ast.cam.ac.uk

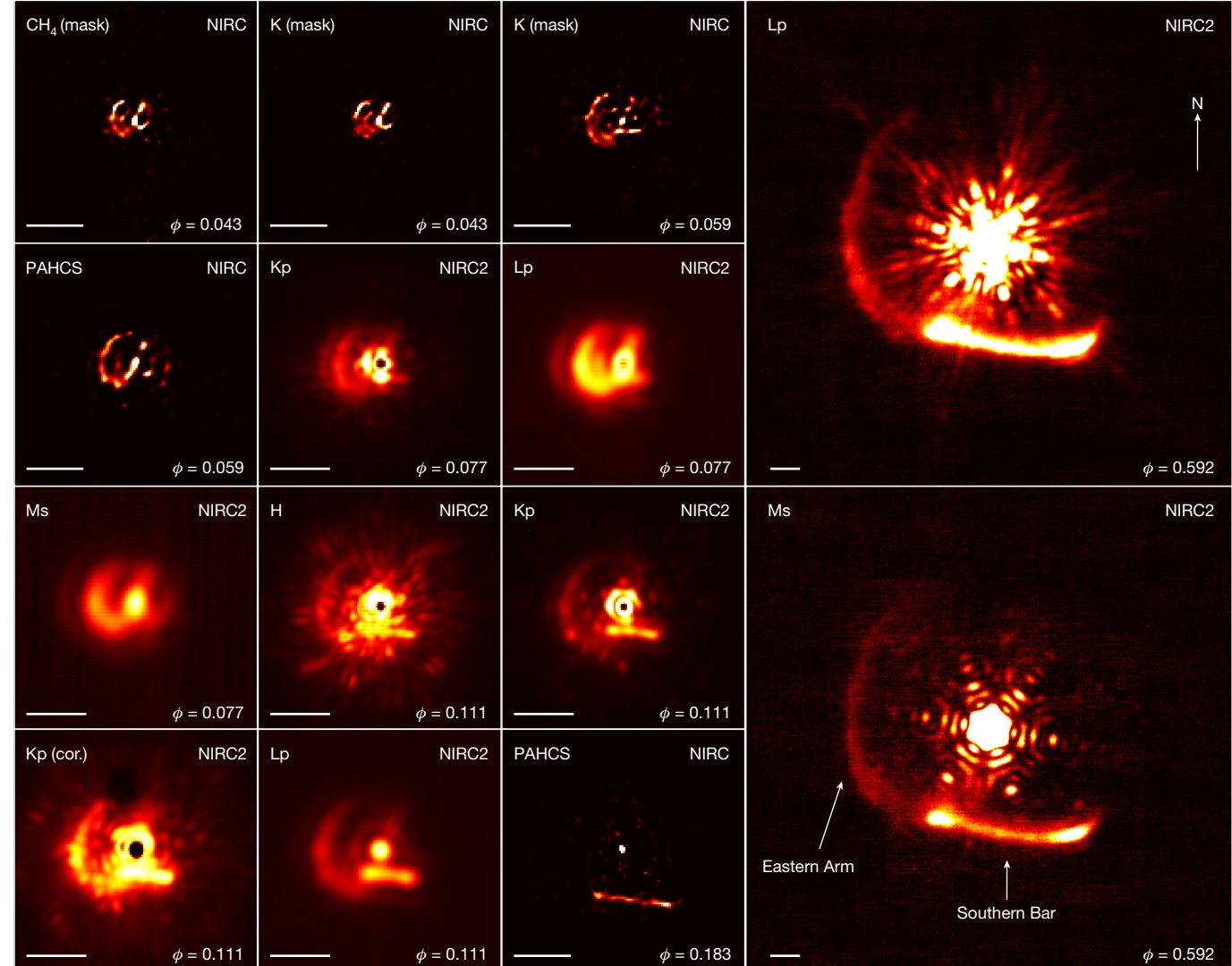

**Fig. 1 | Near-infrared imagery of WR140's expanding circumstellar dust structure.** The observing bands (top left), instrument (top right) and orbital phase $\phi$ (bottom right) are labelled in each panel. Coronagraphic (cor.) and aperture masking (mask) images are labelled in parentheses after the observing bands. All other images were observed with the full aperture. The scale bar in each panel indicates 0.3 arcsec (corresponding to 501 AU at a distance of 1.67 kpc). The images are stretched linearly by the following amounts to accentuate the dust structures that are otherwise faint relative to the bright stellar core: $\phi = 0.077$ Kp-band image stretched between 0 and 0.8 relative to the brightest pixel; $\phi = 0.111$ Kp-band image between 0 and 0.5; $\phi = 0.111$ H-band image between 0 and 0.8; $\phi = 0.592$ Lp-band image between 0 and $1.5 \times 10^{-3}$; $\phi = 0.592$ Ms-band image between $1 \times 10^{-4}$ and $5 \times 10^{-3}$. All other images are not stretched. An observing log is presented in Extended Data Table 1.

to orbital velocity, and deeper astrophysical insight into the plume generative model is required.

We found that two modifications to the geometric model enable it to account for the lack of the north-western outer dust arcs, both of which have physical motivation in the mechanics of the system. First, as dust nucleation and condensation sensitively depend on the physical conditions mediated by stellar winds[3], a change in the shock structure will occur over the orbit, resulting in a dust production rate that varies continuously within the dust-producing segment of the orbit. We found that when the dust production rate is smoothly reduced to a local minimum at periastron in the model, we obtain a significantly improved fit that removes the western dust arc from the image, in accordance with observation.

Second, asymmetries introduced at the wind base can be amplified into asymmetric large-scale structures. Williams et al.[4] found that an increased dust density at the trailing edge of the shock cone relative to the orbital direction produced a better fit than a uniform model. This asymmetry could be introduced by the fast orbital motion near periastron, which results in an intrinsic 'headwind' against the instantaneous

direction of orbit. By allowing dust to preferentially form on the trailing edge of the conical shock front and smoothly decrease in density towards the leading edge, we again obtained an improved fit with the relative brightness of the northern dust structures reduced. A schematic diagram illustrating the two effects is displayed in Fig. 3a.

Figure 2b,c displays model images under the two modifications to the original model (see also Extended Data Fig. 1). We find that a very close fit to the data is achieved when both the orbital and azimuthal modulation of dust production are combined (Fig. 2d–f). Overlaying the outline of the model on the corresponding $\phi = 0.592$ Lp-band observation shows that the geometry predicted by the model excellently reproduces the dust structures and their location (Fig. 2e).

Previous interpretations of dusty WR binaries have focused on identifying the upper limit on the binary separation that enables sufficiently high densities in the colliding winds to form dust. As gas compression is derived from wind–wind collision, it is natural to expect an upper distance threshold above which a sufficiently high density cannot be reached to facilitate dust nucleation. However, our model appears to

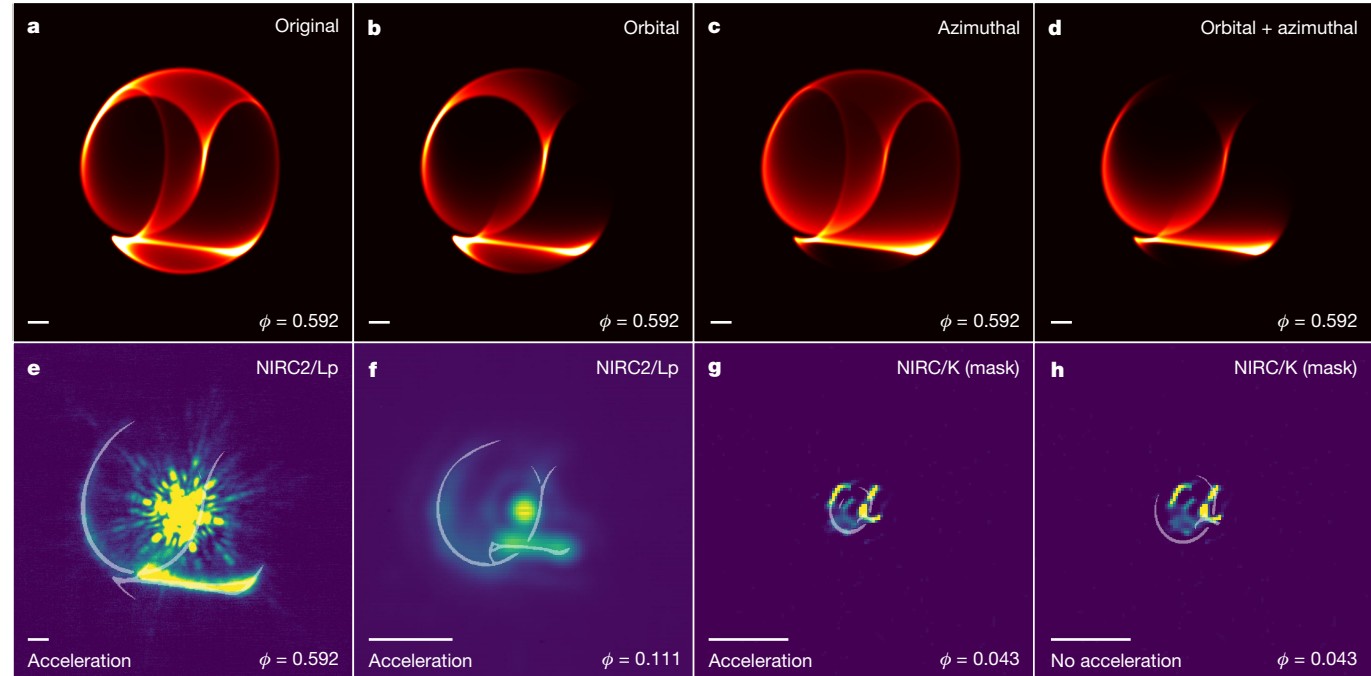

**Fig. 2 | Geometric model of WR140's circumstellar dust structure.**
**a**–**d**, Model images at $\phi = 0.592$ simulated under four different variants of the geometric model. The four models assume that the dust production rate is uniform (**a**), orbitally modulated (**b**), azimuthally asymmetric (**c**) and both orbitally modulated and azimuthally asymmetric (**d**). An animation showing the evolution of the model in time is available in Supplementary Video 1.

**e**–**h**, The outline of the 'orbital + azimuthal' model overlaid on the observations at three different epochs, in which **e**–**g** show models at orbital phases of 0.592, 0.111 and 0.043 adjusted for dust acceleration, whereas **h** shows a model at an orbital phase of 0.043 that is not adjusted for acceleration, resulting in a clear misfit to the data. The scale bar in each panel indicates 0.3 arcsec.

suggest that there may also exist a lower limit, implying that dust is only formed within a Goldilocks zone where just the right conditions of density and temperature are met.

Observationally, several other WR binaries with infrared-imaged dust plumes, such as WR112 (ref. [9]), WR98a[15,16] and WR104 (refs. [5,6]), appear to be continuous dust producers that do not show apparent signs of such a threshold effect. The Apep system appears to oscillate in and out of a single threshold, with dust production occurring near periastron[8]. If dust production in WR140 indeed reaches two maxima over a single orbit as suggested by our model, it would be the first WR binary known to exhibit the full range of this Goldilocks effect.

Previous work by Usov[3] has shown that the degree of WR wind compression and cooling is sensitively dependent on the wind velocity, and so it is plausible that, at very short distances, the WR wind is significantly slowed by its binary companion via radiative braking, thereby reducing the rate of dust production. Lau et al.[17] proposed that this mechanism may be responsible for impeding dust formation in the Gamma Velorum system. Alternatively, near periastron, the wind of the O star may not have accelerated to its terminal velocity before reaching the shock front.

With all other parameters fixed, the final parameter of interest in the model is the dust expansion speed. Dust grains are expected to form in a population of postshock gas where the two stellar winds of the binary collide. This dust may then be accelerated by stellar radiation pressure until reaching the terminal dust drift velocity. The assumption of a constant expansion speed similar to the stellar wind speed has been shown to accurately reproduce the expanding dust shell in similar systems such as WR112 (ref. [9]).

However, the multiepoch observations of WR140 suggest that a uniformly expanding dust shell cannot simultaneously reproduce the time-evolving spatial extent of the dust shell observed in all epochs. Fitting to the $\phi = 0.592$ epoch, which shows the best-resolved dust structures, the model suggests a dust expansion speed of 2,400 ± 100 km s$^{-1}$,

which is broadly consistent with a streaming velocity of 2,170 ± 100 km s$^{-1}$ along the shock cone determined by Fahed et al.[11]. However, extending this model to earlier epochs shows a clear misfit to the location of the dust shell. Figure 2h shows the outline of such a uniform expansion model overlaid on the earliest epoch of imagery at $\phi = 0.043$. The dust shell predicted by uniform expansion fails to fit, being significantly larger than that observed, implying that the average expansion speed up to $\phi = 0.043$ must be lower than in subsequent epochs.

Dropping the assumption of uniform expansion, the locations of the dust structures were fitted independently to each epoch. The resulting expansion speeds yielded a clear accelerating trend, with most of the impulse imparted onto the dust by the first two epochs. Given the uncertainties associated with fitting the location of dust features, direct derivation of the magnitude of acceleration as a function of distance from the star was not possible. To constrain the basic physics, we therefore posited a simple model based on expectations for radiatively accelerated dust as illustrated in Fig. 3b. Dust is not expected to form very close to the star owing to high temperatures and harsh ultraviolet radiation, and so the postshock gas is assumed to originate at a constant drift velocity, $v_0$. At a distance of $r_{nuc}$, dust nucleates and condenses to form an optically thick sheet, which experiences maximal radiation pressure with a constant acceleration, $a_{max}$. The dust continues to expand and eventually becomes optically thin at $r_t$, at which stage the acceleration decreases as $1/r^2$.

Under these working assumptions, we find that dust grains form at a speed of $v_0 = 1,810^{+140}_{-170}$ km s$^{-1}$, which is then accelerated in the optically thick regime by $a_{max} = 900^{+700}_{-400}$ km s$^{-1}$ yr$^{-1}$ up to $r_t = 220^{+150}_{-80}$ AU before becoming optically thin. The dust nucleation radius is fitted to be $r_{nuc} = 50 ± 30$ AU, although this value is not well constrained.

Figure 3c shows the acceleration and velocity as a function of distance resulting from the best-fit model. Radiation pressure has long been suggested to play a major role in accelerating material near WR stars[18]. We find that the best-fit acceleration in the optically thin regime may be

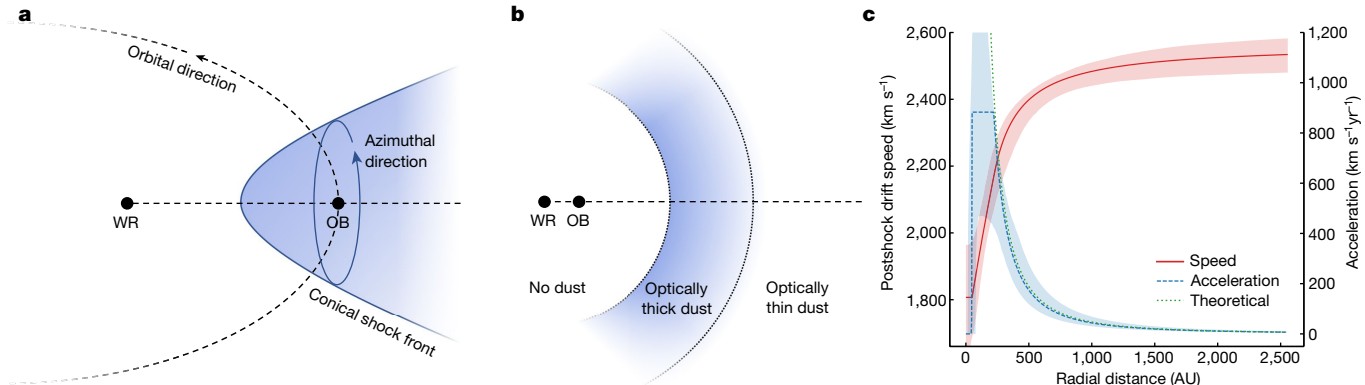

**Fig. 3 | A model for dust production and acceleration in WR140. a**, The 'orbital' and 'azimuthal' directions referred to in this paper relative to the orbit of a binary consisting of a WR star and an O or B star. A dust production rate that varies in the both the orbital and azimuthal directions is found to reproduce the observed structures. **b**, The acceleration regimes near the binary. No dust is produced interior to the dust nucleation radius, where material flows at a constant rate. Upon nucleation, dust forms an optically thick shell accelerated at a uniform rate before the acceleration decreases as the dust becomes optically thin. **c**, The best-fit acceleration (dashed) and resulting dust expansion speed (solid) and the corresponding $1\sigma$ interval of models drawn from the posterior distribution fitted to the data. The theoretically expected acceleration via radiation pressure in the optically thin regime assuming a dust-to-gas mass ratio of 0.02 is overplotted (dotted).

reproduced by radiation-driven acceleration expected from the grain and stellar properties of the system (Methods). It is therefore plausible, if not expected, for radiation pressure to drive dust expansion at the acceleration derived from the observations.

The acceleration of matter by radiation pressure is ubiquitous throughout astrophysics, and is central to a range of astrophysical theatres ranging from the clearing of primordial material in protoplanetary disks around young stars[19] to the formation of planetary nebulae towards the end of stellar evolution[20]. These multiepoch observations of WR140 constitute the first direct detection of dust structures undergoing acceleration in the circumstellar radiation field: more commonly proper motions recorded are under the influence of gravity. Acceleration due to interactions other than gravity, however, are rarely witnessed because typically radiative acceleration zones are very short and terminal velocity rapidly attained. However, as we have discovered, WR binaries produce postshock material that reaches a significant distance from the photosphere before dust condensation, providing a rare laboratory in which the acceleration zone may be probed. This motivates further high angular resolution observations of such dust structures close to their nucleation point to test theories of radiatively driven acceleration physics in the circumstellar environment.

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

## Methods

### Image reduction

Imagery from Keck's NIRC[21] were taken using sparse-aperture interferometry techniques as described in Tuthill et al.[22], with an observing log describing specific configurations given in Extended Data Table 1. After data reduction of sequences of short-exposure data cubes, interleaved with observations of a point-source reference star, our codes deliver calibrated standard interferometric observables (visibilities and closure phases). Image recovery was performed with the BSMEM algorithm[23] at a pixel scale of 10 mas pixel$^{-1}$. The BSMEM algorithm reconstructs a model-independent image iteratively, taking into account both the difference between the model and data ($\chi^2$) to assess the goodness of fit, and the entropy function ($H$) to minimize information and avoid overfitting as the normalized $\chi^2$ converges towards unity. The balance between the two terms during the iterative minimization is set by the regularization weight ($\alpha$), which is adaptively determined by the algorithm. The value of the regularization weight is chosen such that it maximizes the Bayesian evidence (that is, the probability of obtaining the data given the regularization weight) given a Jeffreys prior[24].

Data analysis for the NIRC2 instrument, also operational at the Keck observatory, proceeded by centring and stacking data frames common to the same epoch, observing band and aperture configuration (full pupil or coronagraph) and interpolating outlying pixels in the stacked image. For the $\phi = 0.592$ images that showed the faintest dust structure, we subtracted the sky background determined using the dithered data frames.

### Geometric model

**Original model.** Our model assumes that dust is produced on a thin conical shock front, after which it expands radially away from the binary, generating a spiral form as a consequence of rotation of the nose of the shock with the orientation of the orbiting binary. Noting WR140's well-constrained orbital parameters, we adopted values from Monnier et al.[1] presented in Extended Data Table 2, which were derived using a combination of knowledge from radial velocity[11] and optical interferometry. The model is available at github.com/yinuohan/WR140 (ref. [8]).

Parameters additional to the orbit that are required to define the plume model are the half-opening angle of the conical wind-shock interface ($\theta_w$), the terminal dust velocity and the range of orbital phases over which dust production occurs. We constrained these parameters by fitting the geometry generated by the model to the observed dust structures across the multiepoch images. The images generated by the geometric model assume isothermal dust. This is a reasonable assumption given that the temperature gradient in the orbital direction is expected to be small; however, a more exact model of the emission properties would require further radiative transfer simulations. As the primary objective is to reproduce the geometric structure of the dust plume, the features of interest are the shape and location of dust structures. These properties are difficult to programmatically extract from the observations, thereby hindering the construction of a metric and subsequently using an automated sampling algorithm to explore the parameter space, as found by Han et al.[8] in the case of the Apep system. We therefore fitted the structures manually with an understanding of the physical effects of parameters and visual inspection of the model overlaid on the observations.

**Orbital modulation.** Even though the abrupt turn-on and turn-off are able to closely reproduce corresponding features such as the southern bar and the southern segment of the eastern arm, the orbital modulation of dust production rate once dust production begins appears to require a gradual variation, as suggested by the gradual dimming of the eastern arm towards the north. We modelled the variation of the dust production rate over the orbit as the superposition of two distinct peaks, each of which is modelled as a half-Gaussian function varying over the true anomaly. The turning on and off of dust production corresponds to the peaks of the two half-Gaussians, with dust production gradually suppressed near periastron as shown in Extended Data Fig. 2. We find that a reasonable fit is obtained when the half-Gaussians peak at the same turn-on and turn-off orbital phase as previously fitted, each with a standard deviation of $40^{+30}_{-10}°$ in true anomaly. This results in an interval between half powers of $0.15^{+0.05}_{-0.09}$ yr along the orbit.

**Azimuthal asymmetry.** We assumed that the density of dust being produced at the conical shock front varies over the azimuthal angle about the WR–OB axis as a Gaussian function (with periodic boundary condition). Fixing the peak dust density at the trailing side of the shock cone (where the shock cone intersects the orbital plane), the model was best fitted when the dust density distribution exhibited a standard deviation of $80 \pm 20°$ in the azimuthal angle measured from the trailing edge. This results in a density contrast of $13^{+77}_{-8}$ between the trailing and leading sides of the dust plume.

### Expansion speed fitting

The image of the spiral dust plume is a projection of its true three-dimensional structure, and so measuring the true displacement in three-dimensional space relies on the geometric model.

Different dust features in the same observation are produced at different points in time. The effect of this time offset is particularly important towards earlier orbital phases. Although the construction of the geometric model assumes uniform expansion across the entire dust plume, it is still possible to measure the displacement of specific features within an observation by fitting only to the location of that feature. We therefore measured the displacement of the eastern arm and southern bar independently by fitting to their location separately for each epoch of observation, thereby dropping the assumption of uniform expansion speed. Extended Data Fig. 3 displays a series of model outlines overplotted on the corresponding data for models with and without accounting for acceleration. A misfit between the model-predicted location of features and their observed location in the absence of acceleration, which is particularly prominent at earlier epochs, implies that the inclusion of acceleration is required to explain their motion.

The geometric model predicts the range of orbital phases over which dust is produced, and so it is possible to predict the time when the two dust features were produced for each epoch. Here we assumed that the eastern arm and southern bar are produced at the very beginning and end of a dust production episode, respectively. Subtracting this time point from the time of observation gives the total time since a given feature was produced, giving two sets of displacements as a function of time, one for the eastern arm and one for the southern bar. We then combined the two sets of measurements so that they share the same time axis and used the combined data and a point at the origin for subsequent fitting.

We fitted the model for acceleration driven by radiation pressure described in the main text using a Markov chain Monte Carlo, solving for the differential equations

$$a(r) = \frac{d^2 r}{dt^2} = \begin{cases} 0 & \text{if } 0 \leq r < r_{\text{nuc}} \\ a_{\text{max}} & \text{if } r_{\text{nuc}} \leq r < r_t \\ \dfrac{a_{\text{max}} r_t^2}{r^2} & \text{if } r_t \leq r \end{cases} \qquad (1)$$

at each iteration to obtain $r(t)$, which is then compared with the data. The reported fitted results are derived from the median of the marginalized posterior distributions, with uncertainties estimated using the 16th and 84th percentiles. Extended Data Fig. 4 compares the measured location of dust with the best-fit model to these measurements with and without accounting for acceleration. The systematic offset

in residuals for the model, assuming that the radial distance of dust is proportional to time since production, suggests that uniform expansion is not a suitable model.

We note that acceleration fitting relies on knowledge of the time of production of the dust features being fitted to, which in turn is derived from the orbital phase of the earliest and most recently produced dust, which carries uncertainties. Even though the displacement–time measurements at most orbital phases appear to be approximately linear, and even if we were to drop the assumption that displacement is strictly proportional to time in a model without acceleration, a best-fit straight line fitted through these points does not intercept the origin, implying the presence of acceleration at least at very early stages of the expansion. Such an offset from the origin arises from the combined measurements of both the eastern arm and southern bar, and any systematic errors resulting in such an offset across both sets of measurements is unlikely.

## Acceleration from radiation pressure

To estimate the expected effect of radiation pressure, we assumed stellar and grain properties adopted by the dust grain model by Lau et al. (manuscript in preparation), which includes a combined stellar luminosity of $L_{bin} = 1.43 \times 10^6 L_\odot$, a grain size of $a_g = 0.04$ μm, a grain density of $\rho_g = 1.6$ g cm$^{-3}$ (ref. [25]) and a radiation-pressure efficiency of $Q_{pr} = 1$ as appropriate for hot stellar sources given the relatively large grain size[26]. We calculated the radiation force experienced by an individual dust grain as

$$F_{rad} = Q_{pr} \sigma_g \frac{L_{bin}}{4\pi r^2 c} \tag{2}$$

where $\sigma_g$ is the cross-sectional area of a dust grain, $r$ is the distance of the dust grain from the star and $c$ is the speed of light. Assuming that any gas is comoving with the dust as it accelerates, we found that a dust-to-gas ratio of 0.019 reproduces the best-fit acceleration model derived from the observations.

Lau et al.[17] estimated a mean dust-to-gas ratio of approximately $4 \times 10^{-5}$ in the system by comparing the total dust mass derived from observations with the stellar mass-loss rate; however, the larger dust-to-gas ratio found in this study is not surprising given that dust production exhibits significant spatial and temporal variation. As the geometric model suggests, dust is only produced over a fraction of the orbit, and further orbital modulation implies that the dust features (the eastern arm; the southern bar) probed in this study correspond to only a very small fraction of the orbit. Furthermore, dust production occurs only in a confined stream. The two effects together imply that dust in the system is expected to be highly concentrated in the two structures probed in this study, which may account for the more than two orders of magnitude increase in dust density in these structures relative to the mean density, assuming a uniform dust distribution across the system.

Additional contributions to the acceleration could theoretically include further wind collision downstream, which is not modelled here. However, the two stellar winds rapidly become parallel farther away from the apex of the shock front and so this effect is not expected to be a major contribution given the geometry.

The interval of constant acceleration ($a_{max}$) models a scenario in which the dust is initially optically thick, harvesting all photon momentum available over the area that it spans. The extent to which this assumption may hold true motivates an important point of observational follow-up, such as by monitoring variations in the spectral energy distribution of newly formed dust over time.

## Instrumental and optical depth effects

Critical constraints on the accelerating motion of the dust plume are provided by data at early orbital phases, which were observed with aperture masking interferometry. It is therefore important to ensure that the plate scale of images recovered with this technique is not adversely affected by systematic biases, for example due to different observing bandpasses. Aperture masking observations of reference binary stars of known configuration spanning a diverse range of filters and bandwidths at each epoch (Alp UMi with CH$_4$ and Beta Del with KCont in June 2001; HD 2150123 with H, K in July 2001; Zeta Aqr with H, K, Beta Del with KCont and HD 2150123 with H, K in July 2002) all recovered separations within errors of anticipated locations, which suggest that bandwidth effects do not adversely affect proper motion tracking as performed in this study.

Furthermore, the same apparatus and pipelines have also been applied as part of a relatively well-tested and mature experiment that has been benchmarked, delivering numerous multiepoch astrophysical studies of similar resolved structures in other contexts. Notable science targets whose structures were registered and tracked over these observing epochs include IRC+10216 (ref. [27]) and LkHa 101 (refs. [28,29]). In particular, IRC+10216 had outflow motions tracked by tagging structures in recovered images over many epochs with both broad and narrow filters, with the resulting motions consistent with a physically plausible, monotonically expanding flow, and no biases introduced with bandwidth or other settings[27]. In the WR140 dataset presented in this study, the close agreement between features taken with different filters at the same epoch also argue against plate scale biases being a significant effect.

It is also possible that optical depth effects can alter the apparent spatial extent of structures, as inner dust could potentially shadow outer dust at earlier orbital phases if the dust is optically thick, making the structure appear smaller. However, this is unlikely to alter the observed spatial extent by an amount that could explain the apparent acceleration. If shadowing were to be a prominent effect, the thickness of the dust plume required would result in a significantly different structure at earlier orbital phases that resemble a much smaller half-opening angle (approximately 10° smaller than the best-fit value), and at later orbital phases this would produce thick and fuzzy structures that deviate from the fine structures observed. These observations therefore imply that flux gradients and/or shadowing within a thick dust plume are not large enough to bias our kinematic analysis of WR140.

## Data availability

The NIRC and NIRC2 data underlying this article are publicly available on the Keck Observatory Archive (koa.ipac.caltech.edu/cgi-bin/KOA/nph-KOAlogin) under programme IDs U59N, U2N, U45N, U14N2, U044 and Z273. Source data are provided with this paper.

## Code availability

The dust plume model used in this study is available at github.com/yinuohan/WR140.

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

**Acknowledgements** Y.H., P.G.T. and A.S. acknowledge the traditional owners of the land, the Gadigal people of the Eora Nation, on which the University of Sydney is built and this work was carried out. Y.H. acknowledges funding from a Gates Cambridge Scholarship enabled by Grant No. OPP1144 from the Bill and Melinda Gates Foundation. P.G.T. acknowledges the longstanding support for innovative imaging at the W.M. Keck observatory, and development of the techniques particularly from J. Monnier. The data presented herein were obtained at the W. M. Keck Observatory, which is operated as a scientific partnership among the California Institute of Technology, the University of California and the National Aeronautics and Space Administration. The Observatory was made possible by the financial support of the W. M. Keck Foundation. The authors recognize and acknowledge the significant cultural role and reverence that the summit of Maunakea has always had within the indigenous Hawaiian community. This research made use of NASA's Astrophysics Data System; the emcee package[30]; NUMPY[31]; MATPLOTLIB[32]; and Astropy, a community-developed core Python package for Astronomy[33].

**Author contributions** P.G.T. prepared and performed several sets of observations and reduced NIRC data. Y.H. and P.G.T. devised and fitted the geometric model. Y.H. reduced NIRC2 data, fitted the plume motion and, with input from P.G.T., R.M.L. and A.S., wrote the manuscript. R.M.L. analysed dust properties and interpreted the interaction between dust and stellar winds. A.S. analysed possible optical thickness variations for different grain sizes. All authors contributed to analysing the data and synthesizing interpretations of the physics.

**Competing interests** The authors declare no competing interests.

**Additional information**
**Correspondence and requests for materials** should be addressed to Yinuo Han.

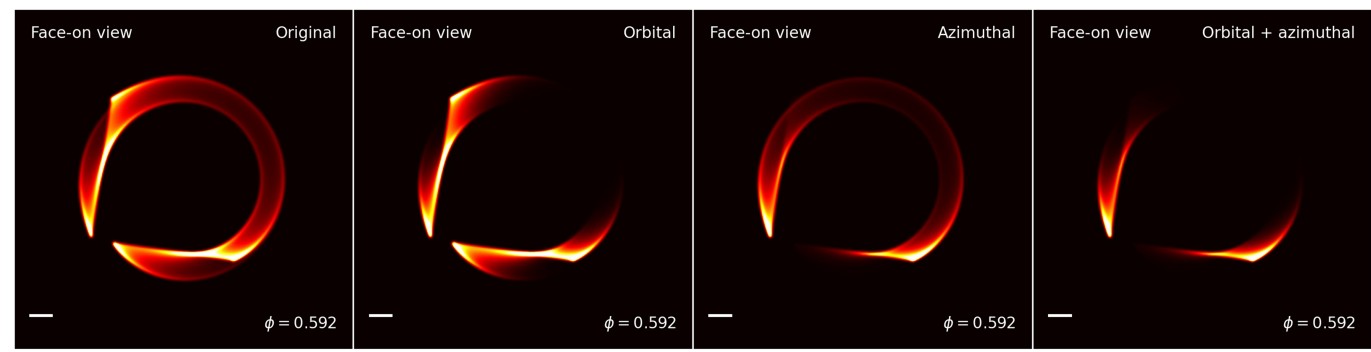

**Extended Data Fig. 1 | Face-on view of the geometric model for WR140.**
The four panels correspond to model images at $\phi = 0.592$ simulated under
the assumption that the dust production rate is uniform (original), orbitally modulated (orbital), azimuthally asymmetric (azimuthal) and both orbitally modulated and azimuthally asymmetric (orbital + azimuthal), respectively.

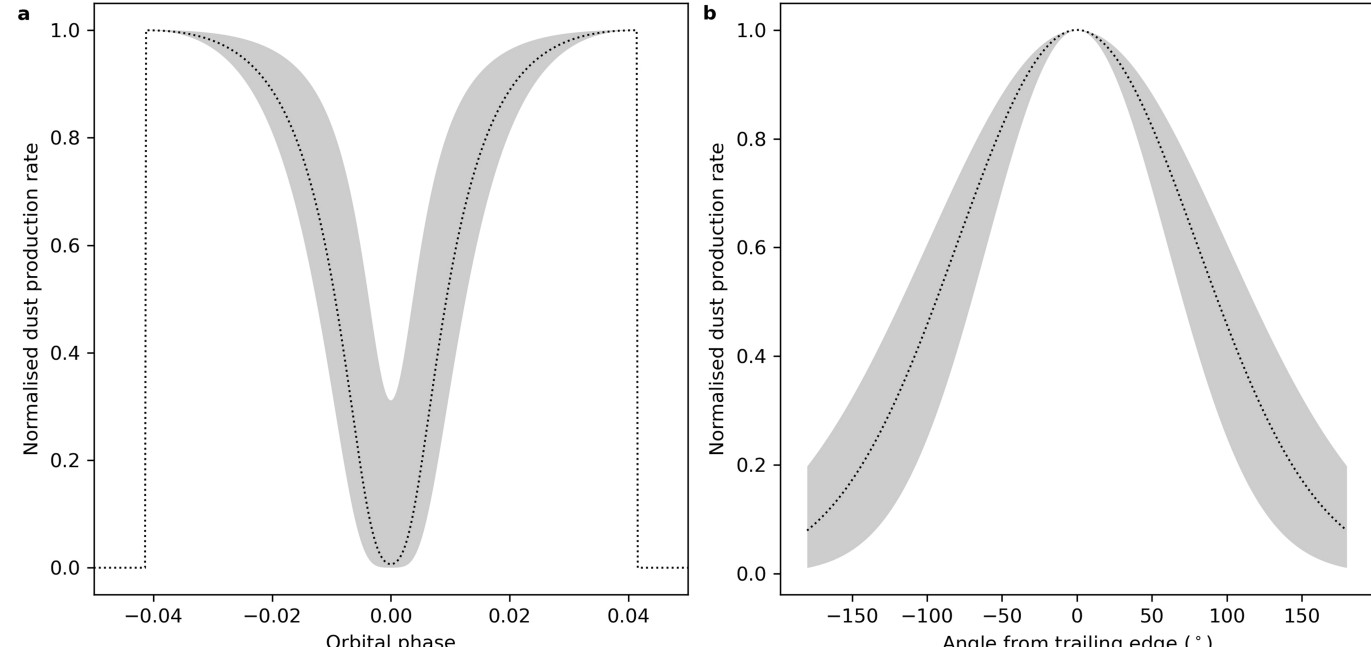

**Extended Data Fig. 2 | Orbital and azimuthal variation of dust production rate in the geometric model.** (a) Dust production rate as a function of orbital phase near periastron. Dust is not produced at all other orbital phases. (b) Dust production rate as a function of azimuthal angle (about the WR–O star axis) relative to the trailing edge. The 1σ uncertainty regions in both plots are shaded.

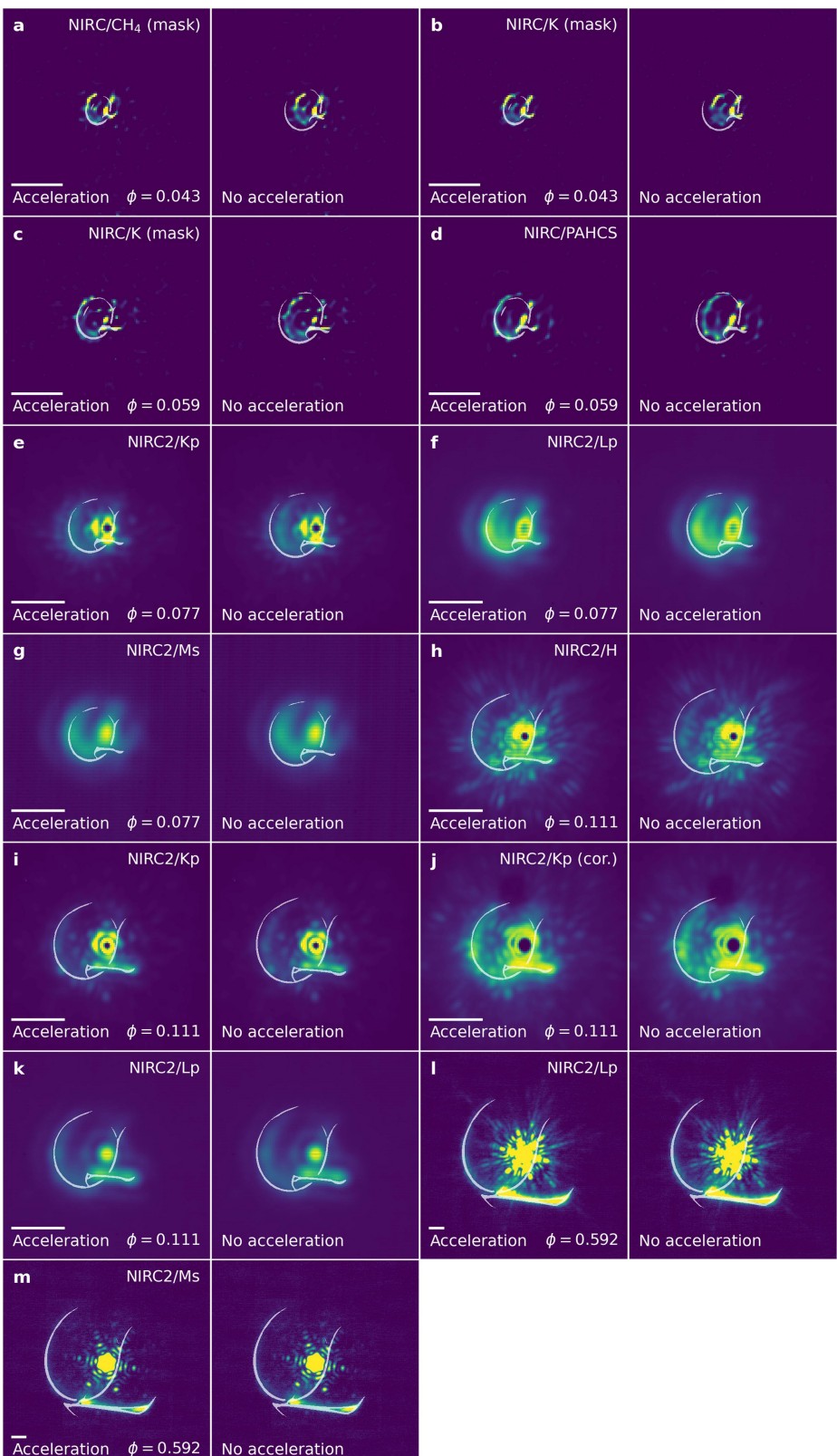

**Extended Data Fig. 3 | Comparison between the model-predicted and observed location of the Eastern Arm for models with and without acceleration.** All observations in which the Eastern Arm is visible are reproduced twice, one of which is overplotted with the outline of the model with acceleration and fitted to the location of the Eastern Arm (left panel of each pair) and the other without acceleration (right panel of each pair). The model without acceleration is fitted to the final epoch of observations. Lack of acceleration results in a misfit at earlier epochs.

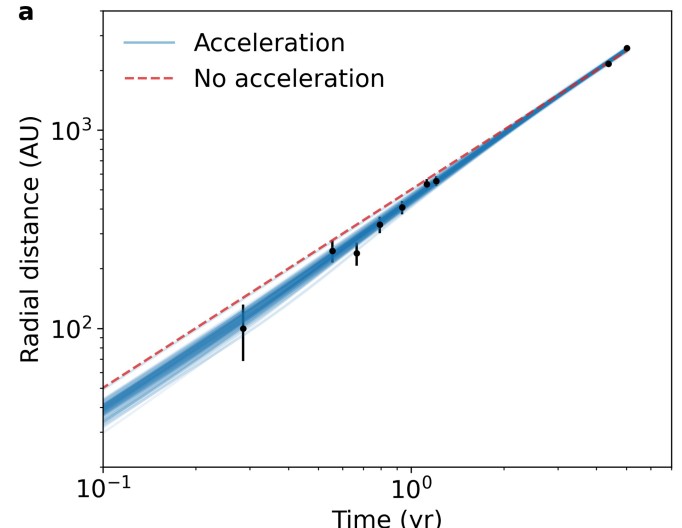

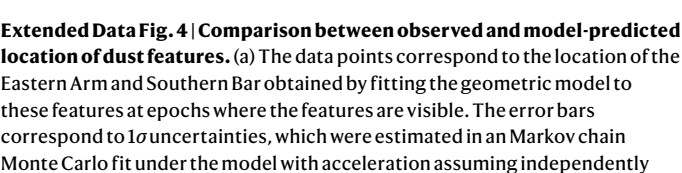

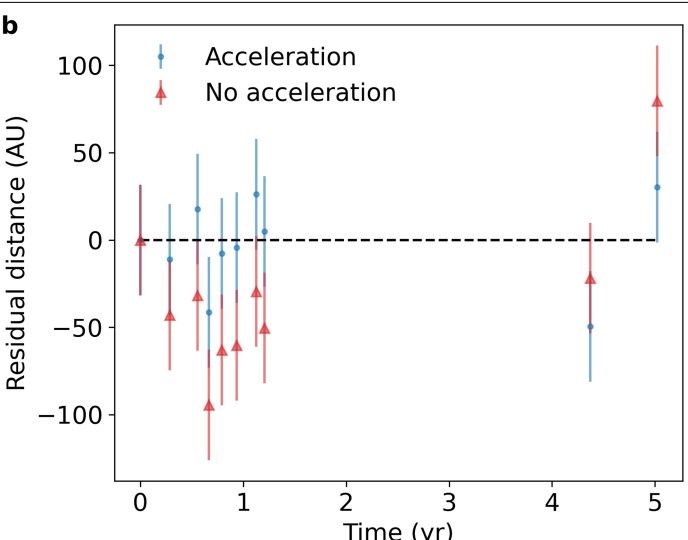

**Extended Data Fig. 4 | Comparison between observed and model-predicted location of dust features.** (a) The data points correspond to the location of the Eastern Arm and Southern Bar obtained by fitting the geometric model to these features at epochs where the features are visible. The error bars correspond to 1σ uncertainties, which were estimated in an Markov chain Monte Carlo fit under the model with acceleration assuming independently and identically distributed Gaussian errors. The solid lines show 100 models randomly drawn from the posterior distribution. The dotted line shows the best-fit model assuming instead that the radial distance of dust features is proportional to time (that is, no acceleration). (b) residuals of the models with and without acceleration. The model without acceleration resulted in a systematic bias in the residuals.

**Extended Data Table 1 | A log of NIRC and NIRC2 observations of WR140 presented in this study**

| Instrument | Filter | Obs Mode | Obs date | Orbital phase |
|---|---|---|---|---|
| NIRC | CH$_4$, K | Masking | 12 Jun 2001 | 0.043 |
| | K | Masking | 30 Jul 2001 | 0.059 |
| | PAHCS | Full pupil | 30 Jul 2001 | 0.059 |
| | PAHCS | Full pupil | 24 Jul 2002 | 0.183 |
| NIRC2 | Lp, Ms | Full pupil | 22 Oct 2005 | 0.592 |
| | Kp, Lp, Ms | Full pupil | 31 Jul 2017 | 0.077 |
| | H, Kp, Lp | Full pupil | 5 Nov 2017 | 0.111 |
| | Kp | Coronagraph200 | 5 Nov 2017 | 0.111 |

Filters [name / central wavelength / fractional bandpass] for the NIRC camera: [CH$_4$ / 2.269 μm / 6.8%]; [K / 2.2135 μm / 19%]; [PAHCS / 3.0825 μm / 3.3%]; and for the NIRC2 camera: [H / 1.633 μm / 18%]; [Kp / 2.124 μm / 16.5%]; [Lp / 3.776 μm / 18.5%]; [Ms / 4.67 μm / 5.2%]. The "Masking" observing mode corresponds to a partially redundant aperture mask as described in Tuthill et al.[22]. "Full pupil" indicates normal imagery, while one epoch also employed an occulting "Coronagraph200" spot. Orbital phases were calculated using orbital parameters derived by Monnier et al.[1].

**Extended Data Table 2 | Properties of the WR140 system**

| | Value | Reference |
|---|---|---|
| Orbital period ($P$) | $2896.35 \pm 0.20$ days | (1) |
| Time of periastron ($T_0$) | $46154.8 \pm 0.8$ MJD | (1) |
| Eccentricity ($e$) | $0.8964^{+0.0004}_{-0.0007}$ | (1) |
| Inclination ($i$) | $119.6 \pm 0.5°$ | (1) |
| Longitude of ascending node ($\Omega$) | $53.6 \pm 0.4°$ | (1) |
| Argument of periastron ($\omega$) | $46.8 \pm 0.4°$ | (1) |
| Semi-major axis ($a$) | $8.82 \pm 0.05$ mas | (1) |
| Distance ($d$) | $1.67 \pm 0.03$ kpc | (1) |
| WR star mass loss rate ($\dot{M}_{WR}$) | $1.7 \times 10^{-5}$ M$\odot$/yr | (2) |
| O star mass loss rate ($\dot{M}_o$) | $3.7 \times 10^{-7}$ M$\odot$/yr | (2) |
| WR star terminal wind speed ($v_{\infty,WR}$) | 2800 km/s | (3) |
| O star terminal wind speed ($v_{\infty,O}$) | 3100 km/s | (3) |
| O/WR momentum ratio | 0.020 | $\dot{M}_o v_{\infty,O}/(\dot{M}_{WR} v_{\infty,WR})$ |

References: (1) Monnier et al.[1], (2) Pollock et al.[34] and (3) Setia Gunawan et al.[35].

**Extended Data Table 3 | Dust production thresholds in the original model**

| | Turn-on | Turn-off |
|---|---|---|
| True anomaly (°) | $-135^{+5}_{-5}$ | $135^{+10}_{-5}$ |
| Separation (AU) | $7.9^{+1.3}_{-1.1}$ | $7.9^{+3.0}_{-1.1}$ |
| Orbital phase | $-0.041^{+0.008}_{-0.011}$ | $0.041^{+0.025}_{-0.008}$ |
| Duration of dust production (yr) | $0.7^{+0.3}_{-0.1}$ | |
| Separation at closest approach (AU) | $1.53^{+0.04}_{-0.05}$ | |

