## [Peer Review File · Nature]

Manuscript Title: Radiation driven acceleration in the expanding WR140 dust shell

Reviewer Comments & Author Rebuttals

Reviewer Reports on the Initial Version:

Referees' comments:

Referee #1 (Remarks to the Author):

The paper presents new, high quality infrared images of the dust cloud made by WR 140 in the 2001 and 2016 periastron passage episodes and uses these to test models for the dust formation process. The WR140 system has become a prototype for its class and has been studied over a wide range of wavelengths, particularly X-ray and infrared. Topically, it is scheduled for observation with the JWST during 2022.

There is a problem with the authors' modelling: they consider dust grains to accelerate from a "standing start", whereas the grains condense within compressed wind formed by collision of the two stars' winds and moving at a substantial fraction of the grains' final velocity. The grains are then accelerated by a few 100 km/s, much less than that in this paper. I have also noted a number of errors, uncertainties and suggestions listed by line number in the paper which need to be addressed. I have also noted instances where the non-specialist reader will need more information.

Nature readers, including most astronomers, need to be told early on that dust formation by WR binaries occurs in a collimated stream originating in a structure formed by the two collision of the winds of the two stars and rotating with their orbital motion, producing pinwheels – or arcs if the dust forms for only part of the orbit.

Line 1: The binary separation of WR140 varies by a factor ~ 18 during the orbit, so the authors need to give a range, not a single value.

Line 37: The Monnier+ 2011 paper gives the orbit.

Line 43: The introduction of a cone will baffle non-specialist readers. They need to be told that the cone is an approximation for the wind collision region at large distances from the stars where the velocity of the compressed wind has reached its asymptotic value. A reference to Usov 1991, Canto+ 1996 or another paper with a sketch would help.

Line 46: Luhrs spells his name with an umlaut in the original paper. Is the change to "ue" to accord with Nature house style?

Line 46: Not infrared excess emission, but 5696A CIII in the optical. In the infrared, the 10830 HeI subpeak was modelled by Williams+ 2021 giving a comparable result, over a larger phase range and with higher S/N.

Line 47: It may be worth saying that Gayley eqn 9 is for radiative post-shock conditions, appropriate here. As Gayley points out, the relation came from the earlier paper by Canto+ 1996, which would be a better reference.

Line 49: The wind momentum ratio needs to be compared with expectations from the wind terminal velocities and mass-loss rates of the stars. Readers need a table giving these and other significant properties of the system, with primary references. This could include the orbital parameters given in the text later.

Line 129: Besides radiative braking of the WC wind, another possible inhibiting effect when the stars are closest is that the wind of the O star may not have had space to accelerate to its terminal velocity.

Line 134: Inhibition of dust formation on the leading side of the shock front may be responsible for the azimuthal asymmetry of dust formation but is not intuitive; does any of the theoretical modelling of colliding winds published since the original Williams+ 2009 suggestion of this dust formation asymmetry shed any more light on this phenomenon?

Lines 141-43: Certainly the newly formed grains are accelerated by radiation pressure, but why would the ram pressure affect the grains any more than it shapes the wind collision region? Writing “terminal wind speed” is confusing: the stars have terminal wind speeds and so does the compressed wind; it would be better to use terminal dust drift velocity – which is determined by the balance of radiative acceleration and supersonic drag – as discussed in earlier references.

Line 153-55: This comparison in Fig. 2 is key to the paper's argument but not watertight as there might be other reasons for the difference. It would be more convincing if the authors plotted the distances of the chosen dust features against time or phase and compared these with one or more models.

Line 174: Is there evidence for an optically thick sheet? If the newly formed dust was optically thick, its SED would be Planckian, which would change to modified Planckian, as the dust thinned assuming its emissivity law followed that of carbon grains.

Line 179: What is the 1760 km/s “gas drift speed”? Is it the compressed wind observed in the CIII and HeI subpeaks? The grain speed must be the sum of the speed of the compressed wind in which it condenses (the terminal or asymptotic value where the region can be approximated by a hollow cone) plus the drift speed from radiation pressure balanced by drag. But Fig 3c shows the grain speed starting from zero, which cannot occur in a system characterised by fast winds.

Line 180: How is the acceleration calculated? How does it compare with that calculated from grain properties (Q_{pr}) and the stellar radiation field?

Line 233: Monnier (2011)?

Line 236: “consistent with assuming” could be rephrased to “assume”

Lines 252: A quantitative duration of the pause, e.g. interval between half-powers, or the half-widths of the Gaussians, is needed.

Lines 252-258: My reading is that the peaks of the half Gaussians coincide with the abrupt turnings on and off but this could be made clearer. The contrast between the abrupt initial turn-on and final turn-off and the gradual pause in the middle requires some comment on possible reasons for the difference.

Line 258: What is meant by “standard deviation of 40^{+30}_{-20} ”.

Line 264: What is meant by “standard deviation of $80 \pm 20^\circ$ in dust density”? What is the angle measured between? What is the density contrast between the leading and trailing sides?

Line 285: When were the dust features referred to produced? Have the Eastern Arm and Southern Bar been identified?

(signed) Peredur Williams

Referee #2 (Remarks to the Author):

This paper describes near-IR observations of the dust spiral in WR140, the canonical colliding wind binary (CWB) episodic dust maker. The observations have been taken over 2 orbital phases, and show phase-repeatability of the detected IR emission. The authors fit a geometrical model of the dust to the data and determine some key parameters in their model. They also claim to detect an acceleration of the dust features, which they posit is caused by radiation pressure on the dust grains.

I found the observations intriguing but the modelling and interpretation lacking. There are many aspects to the modelling that are unclear. Issues are:

- 1) Is the gas and dust assumed to move together?
- 2) Does the dust nucleation start at the apex of the wind collision region (WCR)? Or once the nucleation radius of $r_{\text{nuc}} = 50^{+30}_{-30}$ AU (mentioned on p4) is reached?
- 3) One possible cause of the acceleration which is not noted is from further interaction with the stellar winds. Can this be eliminated as a possibility? The authors note that the dust in n WR112 does not show any sign of any acceleration, but do not explore why one system should show acceleration and the other not.
- 4) What effect might there be on the geometry of the WCR when it dips into the acceleration zone of the winds around periastron? What effect might radiative braking have on the WCR geometry? How

might this affect the observed location of the dust?

5) What effect might instabilities in the WCR have on dust production and acceleration of the dust?

6) One does not expect the properties of the WCR to be identical about periastron. But it seems like the dust production assumes this - ie a symmetrical switch-on/switch-off about periastron. Have the authors examined an asymmetrical orbital production of the dust? The authors also note in the final paragraph of the "Expansion speed fitting" section that "as long as dust production is centred at periastron... a model without acceleration." [is not tenable] - so can a model without acceleration fit the data if the dust production is asymmetrical about the orbit?

7) Williams et al (2009) find the dust production appears to be greatest on the trailing edge, which is also found here. What physical reason can be found for this?

I consider the results to be of interest to astronomers, but not to people outside this discipline. The authors "fit the structures" manually, so it is hard to get a feel for how well constrained the model parameters are, and their accuracy given the many model assumptions. The acceleration model is fitted by a Markov chain Monte Carlo which is appropriate. However, the fit shown in Fig 3c should have some error bars on.

Overall I am not convinced that the authors have a fully consistent model or have explored all options in their model. For this reason I do not recommend publication as the paper currently stands.

Referee #3 (Remarks to the Author):

The paper presents a beautiful dataset of multi-epoch imagery of WR140, showing the expansion of the dust shell around this Wolf-Rayet binary. One finding of potentially high originality and broad significance is the detection of radiation-driven acceleration of the newly-formed dust. I find it plausible that this might be the most direct observational evidence for radiation-driven acceleration to date. I believe further work is needed to demonstrate that this finding is robust, eg with respect to instrumental and optical depth effects (see below).

The abstract is well-written and outlines the broader relevance of the study. Some further streamlining is needed to make it compatible with the Nature formatting guidelines. Also other aspects of the presentation should be improved -- for instance the figures lack descriptions of image orientation and color representation.

--- major comments ---

The robustness of the main conclusion of the paper, namely on radiation-driven acceleration,

depends critically on the observation that the structure measured at the first epoch (phase 0.043) appears somewhat smaller (maybe 10-20%) than predicted by the 'no acceleration' model (bottom-right panel of Figure 2). I think it is important to rule out that observational biases and/or optical depth effects could affect the sizes measured in 2001, especially as these observations were obtained with a different instrument/observing technique than the later observations (aperture masking, while the later epochs used direct imaging):

[1] Interferometric imaging from aperture masking data can be subject to certain biases when using filters with broad bandwidth (such as K band filters in 2001): for strongly resolved objects there could be bandwidth smearing effects that can bias the size of extended structures in the images; for strongly reddened objects, the *effective* central wavelength (i.e. weighted by the numbers of photons per wavelength) can be shifted towards longer wavelengths than the nominal central wavelength of the filter assumed by the image reconstruction algorithm. Both of these effects could result in a systematically smaller structure in the images.

Given that the 2001 images in the CH4 filter and K band filter appear roughly similar in size (Figure 1) I suspect that these potential biases cannot account for a 10-20% size difference, but given the scaling of the images and the absence of visibility plots it is difficult to be certain. I recommend to add a quantity discussion of these effects backed up by a simulation, where synthetic chromatic visibilities are computed from the 'no acceleration' model and then retrieved with the same image reconstruction method as applied to the real data.

Such a simulation would also help to build confidence that the geometry at phase 0.043 / 0.059 is reasonably well represented by the model (important, considering that the model geometry has been tuned primarily to fit the data at phase 0.592) and to identify image artefacts.

[2] You argue that the optical depth of the dust shell decreases as it expands. Thus, with higher optical depth the dust emission from the inner edge of the shell will be dominant, while the surface brightness of the outer shell will be reduced due to shadowing from dust located closer in. Can we rule out that the dust shell has a finite radial extent and appears more compact at the earliest epochs due this effect?

[3] According to the main text (line 160), you fitted the location of the dust structures independently for each epoch. Figure 2 shows the comparison between model and data only for 3 of the 14 measurements. This makes it difficult to assess how well the expansion rate is constrained and how robust the claim of accelerated expansion is. I suggest adding a plot to the method section where the images are overplotted with the model curves for the overall best-fit 'no acceleration' model, and, the model curve as scaled to match the dust structure just at that epoch. This would allow a better assessment of the model residuals.

Also, given that the Eastern arc is present in all images, it would be helpful to plot the separation of the arc (together with some uncertainty estimate, e.g. based on the thickness of the arc and/or angular resolution of the observation) versus orbital phase to illustrate that the expansion does not follow a linear law.

--- minor comments ---

[4] Abstract / bibliography: Please consider citing a few early papers, such as Beals 1929 (MNRAS 90,

202-212), who identified radiation pressure as the likely cause for the expansion of the gaseous shells around WR stars.

[5] In lines 207-209 you explain the use of BSMEM. Please provide more details on the choice of the regularisation parameters and other details, such as pixel scale and χ^2 .

[6] In lines 297-301 you discuss a displacement-time diagram for the Eastern Arm and Southern Bar. Please show this diagram.

[7] Figure 1 and 2 should include a colorbar and information about the image orientation on sky. Also, it took me some time to understand the plots -- consider adding grid lines to separate the individual panels.

[8] The way how panels are referred to from the text is often ambiguous. For instance, "row 1, column 1" assumes that the reader counts top->bottom and left->right, which is not a universal convention worldwide. I suggest to label the panels A, B, C, ...

[9] Table 1 caption: Please clarify -- is the 'unmodified model' identical with the 'Original' model referred to in Figure 2?

Author Rebuttals to Initial Comments:

Referees' comments:

Referee #1 (Remarks to the Author):

The paper presents new, high quality infrared images of the dust cloud made by WR 140 in the 2001 and 2016 periastron passage episodes and uses these to test models for the dust formation process. The WR140 system has become a prototype for its class and has been studied over a wide range of wavelengths, particularly X-ray and infrared. Topically, it is scheduled for observation with the JWST during 2022.

There is a problem with the authors' modelling: they consider dust grains to accelerate from a "standing start", whereas the grains condense within compressed wind formed by collision of the two stars' winds and moving at a substantial fraction of the grains' final velocity. The grains are then accelerated by a few 100 km/s, much less than that in this paper. I have also noted a number of errors, uncertainties and suggestions listed by line number in the paper which need to be addressed. I have also noted instances where the non-specialist reader will need more information.

We thank the referee for pointing out the concern regarding the modelling. In our model, we assumed the dust grains to start accelerating from some initial velocity, instead of from a "standing start". The best-fit value of the starting velocity was 1760 km/s as discussed from lines 172 to 178 in the original manuscript. In Figure 3c, the curve for the expansion speed corresponds to the right-hand side axis labels of the plot, instead of the left-hand side labels which correspond to acceleration. We acknowledge that the text and figure 3c may not have been sufficiently clear and explicit about this and have made corresponding adjustments, hoping to eliminate any chance of confusion.

Nature readers, including most astronomers, need to be told early on that dust formation by WR binaries occurs in a collimated stream originating in a structure formed by the two collision of the winds of the two stars and rotating with their orbital motion, producing pinwheels – or arcs if the dust forms for only part of the orbit.

We agree that more context should be provided have added a brief description in the introduction to explain the mechanism as suggested. We have attempted to keep this brief to conform with the Editor's instructions for the revision but hope that the adjustments to the text have made this point clearer.

Line 1: The binary separation of WR140 varies by a factor ~ 18 during the orbit, so the authors need to give a range, not a single value.

We have adjusted the text to display a range corresponding to the unprojected angular separation at periastron and apastron.

Line 37: The Monnier+ 2011 paper gives the orbit.

We have corrected the typo in the reference to Monnier+2011 in the text.

Line 43: The introduction of a cone will baffle non-specialist readers. They need to be told that the cone is an approximation for the wind collision region at large distances from the stars where the velocity of the compressed wind has reached its asymptotic value. A reference to Usov 1991, Canto+ 1996 or another paper with a sketch would help.

We agree that more context is beneficial here and have expanded on the origin of the "cone" in the text and included a reference to Canto+1996 which includes a figure illustrating the geometry.

Line 46: Luhrs spells his name with an umlaut in the original paper. Is the change to "ue" to accord with Nature house style?

We have adjusted the reference to spell "Lührs" in accord with the original paper.

Line 46: Not infrared excess emission, but 5696A CIII in the optical. In the infrared, the 10830 HeI subpeak was modelled by Williams+ 2021 giving a comparable result, over a larger phase range and with higher S/N.

We have made the suggested correction in the text and added a brief comparison with the cone angle derived by Williams+2021 in the text.

Line 47: It may be worth saying that Gayley eqn 9 is for radiative post-shock conditions, appropriate here. As Gayley points out, the relation came from the earlier paper by Canto+ 1996, which would be a better reference.

We have clarified the applicability of the equation to radiative post-shock conditions in the text and modified the reference as suggested.

Line 49: The wind momentum ratio needs to be compared with expectations from the wind terminal velocities and mass-loss rates of the stars. Readers need a table giving these and other significant properties of the system, with primary references. This could include the orbital parameters given in the text later.

We have now included information on the wind velocities and mass-loss rates in Table E3. This enables a comparison between the wind momentum ratio expected and that based on the cone angle measured in this study as suggested.

Line 129: Besides radiative braking of the WC wind, another possible inhibiting effect when the stars are closest is that the wind of the O star may not have had space to accelerate to its terminal velocity.

We agree that a potentially slower wind at periastron is a possible inhibiting effect and have included this as a potential reason for dust production suppression near periastron in the text.

Line 134: Inhibition of dust formation on the leading side of the shock front may be responsible for the azimuthal asymmetry of dust formation but is not intuitive; does any of the theoretical modelling of colliding winds published since the original Williams+ 2009 suggestion of this dust formation asymmetry shed any more light on this phenomenon?

We are not aware of any subsequent work that explored mechanisms that could cause the asymmetry about the WR-OB axis. One way to consider this might be that the orbital speed near periastron is on the order of 100 km/s, which is a noticeable fraction of the wind speed, giving rise to an intrinsic asymmetry in the configuration. A strong headwind may result in the nose of the colliding-wind region to "bend" towards the direction of orbital motion of the O star at small distances from the star, potentially exposing more material coming from the WR wind towards the trailing side. We have added a brief description on the potential origin of the asymmetry in the text.

Lines 141-43: Certainly the newly formed grains are accelerated by radiation pressure, but why would the ram pressure affect the grains any more than it shapes the wind collision region? Writing "terminal wind speed" is confusing: the stars have terminal wind speeds and so does the compressed wind; it would be better to use terminal dust drift

velocity – which is determined by the balance of radiative acceleration and supersonic drag – as discussed in earlier references.

We agree with the comment and have removed the reference to ram pressure in the statement. We have also replaced “terminal wind speed” with “terminal dust drift velocity” in the text to avoid ambiguity.

Line 153-55: This comparison in Fig. 2 is key to the paper's argument but not watertight as there might be other reasons for the difference. It would be more convincing if the authors plotted the distances of the chosen dust features against time or phase and compared these with one or more models.

To more robustly demonstrate the argument, we have included an additional plot of distance against time, comparing the acceleration model with a no acceleration model. To further demonstrate that a uniform expansion does not fit the data, we have included an extended data figure that overlays the uniform expansion model on the data, which shows a misfit at all but the final epoch (to which the uniform expansion speed was fitted). We have explored variations in all free parameters not directly constrained by the Monnier+2011 orbit, including the cone opening angle, expansion speed and onset and offset phases of dust production and no combination of these parameters was able to reproduce the location of the features at all epochs. However, including an acceleration in the model immediately resolved this issue, producing features in the model that closely align with the data. We have included additional figures E3 and E4 in the extended data section to demonstrate this point.

Line 174: Is there evidence for an optically thick sheet? If the newly formed dust was optically thick, its SED would be Planckian, which would change to modified Planckian, as the dust thinned assuming its emissivity law followed that of carbon grains.

Assuming that the dust starts out with uniform acceleration due to it being optically thick was the simplest model assumption that could avoid unphysically large accelerations in a $1/r^2$ regime for small values of r , allowing us to calculate the momentum harvested from the acceleration. The extent to which this assumption is correct is, as we now note in the paper, an excellent topic for further inquiry. At a level of greater detail, we expect subsequent models to require further complexity. We agree that the shape of the SED would be a great tracer, and this would be an important point for observational follow-up. We have included a brief discussion on this assumption in the methods section.

Line 179: What is the 1760 km/s “gas drift speed”? Is it the compressed wind observed in the CIII and HeI subpeaks? The grain speed must be the sum of the speed of the compressed wind in which it condenses (the terminal or asymptotic value where the region can be approximated by a hollow cone) plus the drift speed from radiation pressure balanced by drag. But Fig 3c shows the grain speed starting from zero, which cannot occur in a system characterised by fast winds.

We acknowledge using “gas drift speed” is ambiguous here and have reworded this in the text. Here, 1760 km/s is the velocity at which dust grains form, just prior to being accelerated, as fitted under our acceleration model. This is therefore the “initial dust drift velocity”. As mentioned earlier in this document, we have adjusted the text and Figure 3c in which the axes may not have been sufficiently clear to clarify that our model does not assume the grain speed to start from zero.

Line 180: How is the acceleration calculated? How does it compare with that calculated from grain properties (Q_{pr}) and the stellar radiation field?

The acceleration reported here was obtained by fitting our simple acceleration model (formation of dust at r_{nuc} , with constant acceleration up to $r_{transition}$, and $1/r^2$ acceleration onwards, as illustrated in Figure 3c) to the measured locations of dust features, as described in more detail in Line 279 to 293 in the original manuscript. We agree that it would be informative to compare the derived acceleration to expectations from the radiation field and the grain properties, and have added a section on this comparison in the text and in Fig 3.

Line 233: Monnier (2011)?

We have corrected the typo in the reference in the text.

Line 236: “consistent with assuming” could be rephrased to “assume”

We have rephrased this in the text as suggested.

Lines 252: A quantitative duration of the pause, e.g. interval between half-powers, or the half-widths of the Gaussians, is needed.

We have added the interval between half-powers in the text as suggested.

Lines 252-258: My reading is that the peaks of the half Gaussians coincide with the abrupt turnings on and off but this could be made clearer. The contrast between the abrupt initial turn-on and final turn-off and the gradual pause in the middle requires some comment on possible reasons for the difference.

We have clarified that the peaks of the half Gaussians coincide with the abrupt turning on and off in the text, and have added a comment on why this may be different from the more gradual suppression near periastron.

Line 258: What is meant by “standard deviation of $40^{+30}_{-20}^\circ$ ”.

Here our fitted standard deviation value with uncertainties means that a half-Gaussian dip in dust production rate with a standard deviation of 40 deg along the orbit best

suppresses the dust in a way that matches the images, but a standard deviation down to 20 deg or up to 70 deg could also feasibly reproduce the observed structure. This relatively large uncertainty means that the exact sharpness of the suppression of dust production near periastron is not very well constrained. We have included extended data figure E2 to outline this point.

Line 264: What is meant by "standard deviation of $80\pm 20^\circ$ in dust density"? What is the angle measured between? What is the density contrast between the leading and trailing sides?

Under our model for the distribution of dust azimuthally around the WR-O axis, the dust density peaks at the trailing edge of the cone, and decreases smoothly further away from the trailing edge in a Gaussian fashion. The standard deviation of 80 ± 20 deg in dust density here refers to the fitted value for the standard deviation of this Gaussian, which is a measure of how sharp this decrease is away from the trailing edge. We have included extended data figure E2 to illustrate this more clearly and included the density contrast between the leading and trailing sides in the text as suggested.

Line 285: When were the dust features referred to produced? Have the Eastern Arm and Southern Bar been identified?

Within one episode of dust production, the Eastern Arm corresponds to the earliest dust produced and the Southern bar the latest. Given that dust production is centred at periastron, the duration of dust production in Table 1 implies that the Eastern arm started being produced 0.33 yr before periastron and the Southern Bar 0.33 yr after periastron. We have now included labels for these features in Fig 1. These features have been identified in Williams+2009 which we have referenced in the text.

(signed) Peredur Williams

Referee #2 (Remarks to the Author):

This paper describes near-IR observations of the dust spiral in WR140, the canonical colliding wind binary (CWB) episodic dust maker. The observations have been taken over 2 orbital phases, and show phase-repeatability of the detected IR emission. The authors fit a geometrical model of the dust to the data and determine some key parameters in their model. They also claim to detect an acceleration of the dust features, which they posit is caused by radiation pressure on the dust grains.

I found the observations intriguing but the modelling and interpretation lacking. There are many aspects to the modelling that are unclear. Issues are:

1) Is the gas and dust assumed to move together?

Here we modelled the simplest scenario in which dust is moving with gas. Under this working assumption, radiation pressure provides the only significant term associated with a post-shock acceleration. We parametrised the acceleration as first being constant and then decaying as $1/r^2$ in the optically thin regime, allowing for an estimate of the magnitude of acceleration experienced by the dust. We have updated the text in the methods section to explicitly mention this assumption, including a calculation to compare with expectations from stellar and grain properties.

We acknowledge that some degree of small relative motion between the dust and gas is allowed, although not to the extent that the dust becomes supersonic. But to take a step back, the geometric model and subsequent acceleration modelling are directly concerned with the dust structures and their motion. The finding that the dust exhibits any observable acceleration at all holds regardless of whether this acceleration is interpreted in the context of there being significant gas drag or not. The way in which gas physically affects dust motion is therefore not directly probed in this experiment.

2) Does the dust nucleation start at the apex of the wind collision region (WCR)? Or once the nucleation radius of $r_{\text{nuc}} = 50+/-30$ AU (mentioned on p4) is reached?

In our model we assumed that dust nucleation starts once the nucleation radius, r_{nuc} , is reached. Before reaching this nucleation radius, post-shock matter is assumed to flow along the cone's surface at constant velocity (i.e., post-shock drift velocity). We have clarified this in the text.

3) One possible cause of the acceleration which is not noted is from further interaction with the stellar winds. Can this be eliminated as a possibility? The authors note that the dust in n WR112 does not show any sign of any acceleration, but do not explore why one system should show acceleration and the other not.

We modelled the acceleration as purely due to radiation pressure, but residual acceleration due to collisional processes in two volumes of wind may also be affecting the gas. However, downstream from the apex of the wind-collision region, the two winds coming from the two stars rapidly approach a scenario in which they move almost in parallel. The interaction cross-section therefore goes down rapidly further from the apex, i.e. the further downstream, the more the volumes of gas are moving in parallel, and the less interaction is possible for one volume of gas to push another. Further acceleration due to collision of wind down-stream from the apex is therefore not expected to be a major contributor given this geometry. We have included a brief discussion of this point in the methods section.

WR112 continuously produces dust (due to its relatively circular orbit) making it more problematic to tag discrete elements in the flow for proper motion study. However in principle a series of very high resolution observations at short intervals may be able to recover acceleration. The structures of WR112 probed in the mid-IR (Lau+2020) at lower resolution than here are beyond the corresponding (~ 500 au) acceleration region identified and should show uniform velocity, as the model fits confirm.

The precision enabled by the high-resolution imaging presented in this study is therefore unique, offering the opportunity to directly probe the motion of newly formed dust. Of the many astrophysical contexts in which radiation pressure is believed to play an integral role, WR binaries are the only systems we are aware of so far that predictably produces observable and traceable new dust near stars with sufficient speed and density to be observable in a proper motion study. As our study has demonstrated, this can be observationally leveraged to study the effect of radiation pressure on dust. Searching for acceleration in other WR binaries such as WR112 may be an important motivator for future follow-up observations.

4) What effect might there be on the geometry of the WCR when it dips into the acceleration zone of the winds around periastron? What effect might radiative braking have on the WCR geometry? How might this affect the observed location of the dust?

If the binary dips into the acceleration zone of the winds at periastron, or if radiative braking is present, the wind speed might vary across the orbit, but the two winds should vary together. The two effects apply to both winds and so if one wind is slower, the other should also be slower which mitigates the effect on the momentum ratio which determines the opening angle. Of course the wind momentum could still exhibit some variation, albeit likely relatively small.

However, our model posits orbital modulation: suppression of dust production at periastron with dust production rates at ingress/egress more than an order of magnitude greater than at periastron, which is required to match our observations. The mechanism tuning dust production may indeed relate to exactly the physics that the reviewer identifies. However given almost all dust nucleates at ingress and egress, the structure is dominated by these two intervals. The interval of suppressed production in between is

almost invisible in the data. So in the event a significantly different cone angle occurs when the binary dips into the acceleration zone around periastron (as this review speculates) this would not result in an observable difference since little dust traces this part of the orbit. The dust structures that are recovered are therefore largely unaffected by potential variations in the cone angle near periastron.

The model we have presented, incorporating spherical outflow and modulated dust production efficiency, produces exquisite fits to the structure and remains the simplest possible model to achieve such an outcome. Although additional model complexity may provide more insight into the physical processes at play, the data unfortunately cannot support these additional extra degrees of freedom and their incorporation and values would therefore be speculative. Considering the small range of orbital phases in which dust is produced, we therefore consider the simple model with a constant cone angle as appropriate in the context of obtaining radial location measurements of features in the data.

5) What effect might instabilities in the WCR have on dust production and acceleration of the dust?

It is possible that radiative instabilities could alter cooling and thus the dust production rate at tens of AU. This could be related to orbital modulation of dust production, in which dust production is found to be suppressed near periastron. However, we do not expect instabilities to play a major role on acceleration out at several hundreds of AU where we find acceleration to occur. Instabilities could also cause the resulting dust structure to locally deviate from the smooth conical surface used to model the large-scale structure, however as we are only concerned with the large-scale geometry of the system, we don't expect instabilities to be a major factor. For example, in the NIRC2 images, the Eastern Arm appears to be slightly clumpy, but that does not prevent the model from accurately reproducing the smooth, large-scale geometry of this structure. It is also pertinent to point out the very high degree of cycle-to-cycle repeatability (in both photometry and imaging) exhibited by this system, arguing against stochastic instability being a large factor. Despite interesting contributors to the detailed structure and variations in dust production rate, the overarching theme remains that a simple model of the geometry could be used to locate dust structures which appear to have accelerated following their production, the most plausible explanation of which is due to radiation pressure.

6) One does not expect the properties of the WCR to be identical about periastron. But it seems like the dust production assumes this - ie a symmetrical switch-on/switch-off about periastron. Have the authors examined an asymmetrical orbital production of the dust? The authors also note in the final paragraph of the "Expansion speed fitting" section that "as long as dust production is centred at periastron... a model without acceleration." [is not tenable] - so can a model without acceleration fit the data if the dust production is asymmetrical about the orbit?

We agree that very clear assumptions on the dust production are important to this paper. Our model did not assume that dust production is symmetric about periastron. Upon fitting to the turn-on and turn-off orbital phases independently, dust production was indeed found to be centred at periastron within uncertainties. We have explored different asymmetric distributions during fitting but those do not produce the correct geometry. We have clarified this in the text to avoid any confusion.

7) Williams et al (2009) find the dust production appears to be greatest on the trailing edge, which is also found here. What physical reason can be found for this?

The orbital speed near periastron is on the order of 100 km/s, which is a noticeable fraction of the wind speed, and particularly so as we already know that dust production behaves as a critical "threshold" phenomenon in this system. It is possible that a strong headwind may result in the nose of the colliding-wind region to "bend" towards the direction of orbital motion of the O star at small distances from the star, giving rise to an intrinsic asymmetry in the configuration such as by sweeping more material coming from the WR wind towards the trailing side. We have added a brief discussion on this potential origin of the asymmetry in the text.

I consider the results to be of interest to astronomers, but not to people outside this discipline. The authors "fit the structures" manually, so it is hard to get a feel for how well constrained the model parameters are, and their accuracy given the many model assumptions. The acceleration model is fitted by a Markov chain Monte Carlo which is appropriate. However, the fit shown in Fig 3c should have some error bars on.

Overall I am not convinced that the authors have a fully consistent model or have explored all options in their model. For this reason I do not recommend publication as the paper currently stands.

We have modified Figure 3c to include uncertainty regions as suggested. Our original cover letter attempted to convey the wider interest in the first direct witness to a celestial body accelerating under radiation pressure; all judgements on merit by us and the referee are to some degree subjective. We believe the comments have helped strengthen the findings presented in the paper and hope that our updated manuscript and the specific responses to the comments provided here have addressed the questions raised by the referee.

Referee #3 (Remarks to the Author):

The paper presents a beautiful dataset of multi-epoch imagery of WR140, showing the expansion of the dust shell around this Wolf-Rayet binary. One finding of potentially high originality and broad significance is the detection of radiation-driven acceleration of the newly-formed dust. I find it plausible that this might be the most direct observational evidence for radiation-driven acceleration to date. I believe further work is needed to demonstrate that this finding is robust, eg with respect to instrumental and optical depth effects (see below).

The abstract is well-written and outlines the broader relevance of the study. Some further streamlining is needed to make it compatible with the Nature formatting guidelines. Also other aspects of the presentation should be improved -- for instance the figures lack descriptions of image orientation and color representation.

--- major comments ---

The robustness of the main conclusion of the paper, namely on radiation-driven acceleration, depends critically on the observation that the structure measured at the first epoch (phase 0.043) appears somewhat smaller (maybe 10-20%) than predicted by the 'no acceleration' model (bottom-right panel of Figure 2). I think it is important to rule out that observational biases and/or optical depth effects could affect the sizes measured in 2001, especially as these observations were obtained with a different instrument/observing technique than the later observations (aperture masking, while the later epochs used direct imaging):

[1] Interferometric imaging from aperture masking data can be subject to certain biases when using filters with broad bandwidth (such as K band filters in 2001): for strongly resolved objects there could be bandwidth smearing effects that can bias the size of extended structures in the images; for strongly reddened objects, the *effective* central wavelength (i.e. weighted by the numbers of photons per wavelength) can be shifted towards longer wavelengths than the nominal central wavelength of the filter assumed by the image reconstruction algorithm. Both of these effects could result in a systematically smaller structure in the images.

Given that the 2001 images in the CH4 filter and K band filter appear roughly similar in size (Figure 1) I suspect that these potential biases cannot account for a 10-20% size difference, but given the scaling of the images and the absence of visibility plots it is difficult to be certain. I recommend to add a quantity discussion of these effects backed up by a simulation, where synthetic chromatic visibilities are computed from the 'no acceleration' model and then retrieved with the same image reconstruction method as applied to the real data.

Such a simulation would also help to build confidence that the geometry at phase 0.043 / 0.059 is reasonably well represented by the model (important, considering that the model geometry has been tuned primarily to fit the data at phase 0.592) and to identify

image artefacts.

We agree with the referee on the criticality of the scale of the masking data, and therefore the need to rule out the potential for systematic biases to creep in. We concur with the logic of both possible issues noted by the referee, yet we also agree that the apparent agreement between the ~7% CH₄ filter and the ~19% K filter to within 5% at the earliest epoch argues against this being a strong effect. This has precipitated additional paragraphs of discussion in the methods section.

Our first (and perhaps ultimately most trustworthy) method of ensuring that the astrometric plate scale in the recovered imagery is robust within the stated errors and can be relied upon while observing bandwidths and other configurations are swapped in and out, is that these results come from a relatively well-tested and mature experiment that has been benchmarked, delivering numerous multi-epoch astrophysical studies of similar resolved structures in other contexts. Perhaps most straightforwardly the plate scale is verified by observation of a reference binary star of known configuration at each epoch. For the three epochs of masking data discussed here, binary stars observed were respectively

1. June 2001 - Alp UMi@ch₄; Beta Del@KCont
2. July 2001 – HD 2150123@H, K
3. July 2002 – Zeta Aqr@H,K; Beta Del@KCont; HD 2150123@H, K

Noting that these binaries span a range of observing configuration and filter bandwidth, all recovered separations were found to lie within errors of anticipated locations.

In addition, and perhaps placing more stringent tests on the full telescope-to-science imaging pipeline for objects of more complex morphology, similar structures (complex circumstellar dust plumes) to WR140 were observed with the same apparatus on the same observing runs, and had images produced using the same data pipelines. Notable science targets whose structures (and multi-epoch motions) were registered and tracked over these observing epochs include IRC+10216 and LkHa 101 (among many others). In particular, IRC+10216 had outflow motions tracked by tagging structures in recovered images over many epochs (with diverse observing settings, both broad and narrow filters etc.), with the resulting motions consistent with a physically plausible, monotonically expanding flow, and no biases introduced with bandwidth or other settings as reported in Tuthill, Monnier and Danchi 2005 (<https://ui.adsabs.harvard.edu/abs/2005ApJ...624..352T/abstract>).

Similarly consistent plate scales and time evolutions were recovered for LkHa 101 (a complex scene including a binary companion) by Tuthill et al. 2002 (<https://ui.adsabs.harvard.edu/abs/2002ApJ...577..826T/abstract>) and Tuthill et al. 2001 (<https://ui.adsabs.harvard.edu/abs/2001Natur.409.1012T/abstract>).

Finally, we also undertook a short study (as proposed by the referee) of the systematics introduced by simulating the effects of varying bandwidth, and the impact of imperfect knowledge of the effective centre wavelength (due to spectral slopes across a wide

observing band). This was done for simple binary star configurations, and the outcomes are further entangled with the detailed way the sampling is done in extraction of the data – making a full report on this a lengthy affair and, in our estimation (given the public published evidence above) not worthwhile to add in full to the supplementary material in the paper. Our outcome was that the bandwidth smearing effect noted by the referee is generally small and can be discarded, for our tested range of simulations, compared to the direct effect of mis-estimation of the centre wavelength. Here the outcomes were exactly as one would expect with error in map positions tracking in proportion to the error in assumed λ_{eff} . Dust at 1500 Kelvin (~around expectations at formation) exhibits a peak in the photon emissivity rate almost exactly in the middle of K-band, and so our starting point here is of an optimally flat flux distribution in this particular part of the infrared. (Extreme) variations in blackbody temperature by +1000/-1000 Kelvin from this value cause a shift in the photon weighted centre of the NIRC full K filter (assumed 2.21 microns, 19% bandwidth) to 2.195 / 2.250 microns from this value: a full shift of only 3% in effective wavelength. Our simulated binary maps over the NIRC K-band tracked this expected shift. In conclusion, we believe our proper motion study of embedded structures in the outflow is not adversely affected by the bandwidth issues that the reviewer has identified, and we include discussion of this important point in the methods section.

[2] You argue that the optical depth of the dust shell decreases as it expands. Thus, with higher optical depth the dust emission from the inner edge of the shell will be dominant, while the surface brightness of the outer shell will be reduced due to shadowing from dust located closer in. Can we rule out that the dust shell has a finite radial extent and appears more compact at the earliest epochs due this effect?

Theoretical models of the colliding-wind dust shell have characterised it with a half-opening angle, θ_w , and indeed also a thickness, $\Delta\theta_w$ (Williams 2009). However, the value of $\Delta\theta_w$ may be in fact be very small. Geometric modelling of the WR binary Apep, for which we have observations of very intricate structures, revealed that a thin sheet produces almost perfect recovery of its structures (Han 2020). This dust was produced decades ago and is expected to be optically thin, and so shadowing is not expected to occur in this system. This implies that the dust plume is indeed likely very thin in that system.

More specifically for WR140, if shadowing were to be a significant effect, i.e., the dust shell is spatially thick and optically thick before becoming optically thin, then at early orbital phases the emission would originate mainly from the inner edge, which would correspond more closely to emission from a thin sheet with a half-opening angle of $\theta_w - \Delta\theta_w$. At later orbital phases, as the dust becomes optically thin, we would then expect to see the full, thick dust shell. Given the orientation of WR140, the projected spatial extent of the structure is not particularly sensitive to the cone opening angle. At the location of the Eastern Arm, for shadowing to decrease the spatial extent by 10% at the earliest orbital phase, $\Delta\theta_w$ must be greater than 10 deg. However, at this much smaller opening angle, the geometry becomes significantly

different from the observed structures. Furthermore, simulating a spatially thick dust shell that is optically thin (modelled by stacking thin sheets with a range of opening angles) produces a very fuzzy structure that significantly deviates from the observed fine structure at the latest orbital phase. The models trialled to explore this idea could therefore not be made to match to single-epoch data.

Both findings point to the conclusion that the observed structure is inconsistent with a spatially thick dust shell and rule out the possibility that shadowing could have caused the emission at earlier orbital phases to appear more compact. We have included additional discussion on this important point in the methods section of the text.

[3] According to the main text (line 160), you fitted the location of the dust structures independently for each epoch. Figure 2 shows the comparison between model and data only for 3 of the 14 measurements. This makes it difficult to assess how well the expansion rate is constrained and how robust the claim of accelerated expansion is. I suggest adding a plot to the method section where the images are overplotted with the model curves for the overall best-fit 'no acceleration' model, and, the model curve as scaled to match the dust structure just at that epoch. This would allow a better assessment of the model residuals.

Also, given that the Eastern arc is present in all images, it would be helpful to plot the separation of the arc (together with some uncertainty estimate, e.g. based on the thickness of the arc and/or angular resolution of the observation) versus orbital phase to illustrate that the expansion does not follow a linear law.

We agree with the suggestion to include the additional plots to demonstrate the presence of significant residuals without accounting for acceleration and have included these in the extended data figures E3 and E4. These additional figures also demonstrate that the inclusion of acceleration results in model outlines that closely align with the data, whereas residuals without acceleration are significant across earlier epochs. We have also included a plot of the displacement of dust structures as a function of time to further demonstrate the need to include acceleration in the model.

--- minor comments ---

[4] Abstract / bibliography: Please consider citing a few early papers, such as Beals 1929 (MNRAS 90, 202-212), who identified radiation pressure as the likely cause for the expansion of the gaseous shells around WR stars.

We agree with the need to cite earlier papers in the field and have included additional references to earlier papers as suggested.

[5] In lines 207-209 you explain the use of BSMEM. Please provide more details on the choice of the regularisation parameters and other details, such as pixel scale and χ^2 .

We have included more details on the use of BSMEM in the methods section.

[6] In lines 297-301 you discuss a displacement-time diagram for the Eastern Arm and Southern Bar. Please show this diagram.

We agree that this is an important plot to display and have included this diagram in Extended Data Figure 4 in the updated manuscript.

[7] Figure 1 and 2 should include a colorbar and information about the image orientation on sky. Also, it took me some time to understand the plots -- consider adding grid lines to separate the individual panels.

We have modified the figures to include only the most important panels and have made the modifications as suggested.

[8] The way how panels are referred to from the text is often ambiguous. For instance, "row 1, column 1" assumes that the reader counts top->bottom and left->right, which is not a universal convention worldwide. I suggest to label the panels A, B, C, ...

We agree with the referee on this suggestion and have modified the labels in the figure as suggested.

[9] Table 1 caption: Please clarify -- is the 'unmodified model' identical with the 'Original' model referred to in Figure 2?

The 'unmodified model' is identical to the 'original model'. We have reworded this in the text to avoid any confusion.

Reviewer Reports on the First Revision:

Referees' comments:

Referee #1 (Remarks to the Author):

I welcome the authors' detailed response to my queries and suggestions and the significant revision of their paper. I am satisfied that the results do demonstrate that the grains are accelerated soon after their formation and that this is the first demonstration of this effect. Within the limited space, the authors have successfully given the context of the phenomenon needed for non-specialist readers.

I am content to recommend the paper for publication.

Referee #2 (Remarks to the Author):

The authors have significantly revised the manuscript and I am happy with the changes that they have made.

Referee #3 (Remarks to the Author):

I appreciate the changes the authors made to the manuscript, in particular to the figures. I feel they addressed my main comments and concerns about the reliability of the results appropriately. I think the author's claim of a 'first direct kinematic detection of dust motion under acceleration by radiation pressure' is reasonably well supported. Of course, it is at the discretion of the Editors whether this discovery is of sufficiently broad interest for non-astronomers to warrant publication in Nature. Below I outline some minor comments on the revised version of the manuscript.

*** minor comments:

[1] I have the impression that tables are not introduced in the order they appear in the text (table E3 before table E2, etc).

[2] line 32/33: avoid complicated structure in sentences (" , which ..., which ...")

[3] In the 'Image reduction' section, the authors added some details on the use of BSMEM. However, the description is still very vague. For instance, it mentions that a maximum entropy regulariser has been used, but does not outline how the regularisation weight has been chosen. In some cases, this can be a critical free parameter in the image retrieval process.

Author Rebuttals to First Revision:

Referees' comments:

Referee #1 (Remarks to the Author):

I welcome the authors' detailed response to my queries and suggestions and the significant revision of their paper. I am satisfied that the results do demonstrate that the grains are accelerated soon after their formation and that this is the first demonstration of this effect. Within the limited space, the authors have successfully given the context of the phenomenon needed for non-specialist readers.

I am content to recommend the paper for publication.

We are glad that the referee is satisfied with our edits and are grateful for the comments which have helped improve the scientific accuracy and clarity of the manuscript.

Referee #2 (Remarks to the Author):

The authors have significantly revised the manuscript and I am happy with the changes that they have made.

We are glad that the referee is satisfied with our edits and are grateful for the comments which have helped improve the scientific accuracy and clarity of the manuscript.

Referee #3 (Remarks to the Author):

I appreciate the changes the authors made to the manuscript, in particular to the figures. I feel they addressed my main comments and concerns about the reliability of the results appropriately. I think the author's claim of a 'first direct kinematic detection of dust motion under acceleration by

radiation pressure' is reasonably well supported. Of course, it is at the discretion of the Editors whether this discovery is of sufficiently broad interest for non-astronomers to warrant publication in Nature. Below I outline some minor comments on the revised version of the manuscript.

We are glad that the referee is satisfied with our edits and are grateful for the comments which have helped improve the scientific accuracy and clarity of the manuscript.

*** minor comments:

[1] I have the impression that tables are not introduced in the order they appear in the text (table E3 before table E2, etc).

We have re-ordered Tables E2 and E3 such that they appear in the order that they are introduced in the text.

[2] line 32/33: avoid complicated structure in sentences (" , which ..., which ...")

We have simplified the structure of this sentence in the text.

[3] In the 'Image reduction' section, the authors added some details on the use of BSMEM. However, the description is still very vague. For instance, it mentions that a maximum entropy regulariser has been used, but does not outline how the regularisation weight has been chosen. In some cases, this can be a critical free parameter in the image retrieval process.

We have included additional detail in the manuscript on regularisation with BSMEM. The algorithm minimises the criterion, $J = \chi^2 - \alpha * H$, in which the regularisation function (H) is taken to be an entropy function to minimise information in the reconstructed image. The regularisation weight (alpha) is chosen adaptively by the algorithm. It uses a Jeffreys prior on alpha and calculates the Bayesian evidence, $P(\text{Data}|\alpha)$, to determine the most likely alpha and update its value as the reconstruction progresses. Details of the minimisation criterion and regularisation function are provided by Baron and Young 2008 (<https://ui.adsabs.harvard.edu/abs/2008SPIE.7013E..3XB/abstract>) which we have now referenced in the text.